# Reproducibility of 2D and 3D Ramus Height Measurements in Facial Asymmetry

**DOI:** 10.3390/jpm12071181

**Published:** 2022-07-20

**Authors:** Nicolaas B. van Bakelen, Jasper W. van der Graaf, Joep Kraeima, Frederik K. L. Spijkervet

**Affiliations:** Department of Oral and Maxillofacial Surgery, University Medical Center Groningen, University of Groningen, 9700 RB Groningen, The Netherlands; jasper.vandergraaf@radboudumc.nl (J.W.v.d.G.); j.kraeima@umcg.nl (J.K.); f.k.l.spijkervet@umcg.nl (F.K.L.S.)

**Keywords:** precision, condylar resection, unilateral condylar hyperplasia, hemimandibular hyperplasia, hemimandibular elongation, cone-beam computed tomography, panoramic radiography, imaging, 3D virtual surgical planning

## Abstract

In our clinic, the current preferred primary treatment regime for unilateral condylar hyperactivity is a proportional condylectomy in order to prevent secondary orthognathic surgery. Until recently, to determine the indicated size of reduction during surgery, we used a ‘panorex-free-hand’ method to measure the difference between left and right ramus heights. The problem encountered with this method was that our TMJ surgeons measured differences in the amount to resect during surgery. Other 2D and 3D method comparisons were unavailable. The aim of this study was to determine the most reproducible ramus height measuring method. Differences in left/right ramus height were measured in 32 patients using three methods: one 3D and two 2D. The inter- and intra-observer reliabilities were determined for each method. All methods showed excellent intra-observer reliability (ICC > 0.9). Excellent inter-observer reliability was also attained with the panorex-bisection method (ICC > 0.9), while the CBCT and panorex-free-hand gave good results (0.75 < ICC < 0.9). However, the lower boundary of the 95% CI (0.06–0.97) of the inter-observer reliability regarding the panorex-free-hand was poor. Therefore, we discourage the use of the panorex-free-hand method to measure ramus height differences in clinical practice. The panorex-bisection method was the most reproducible method. When planning a proportional condylectomy, we advise applying the panorex-bisection method or using an optimized 3D-measuring method.

## 1. Introduction

Unilateral condylar hyperactivity (UCH) is a growth disorder which often results in an asymmetrical presentation of the mandible. Obwegeser and Makek (1986) reported a classification system to differentiate between hemimandibular hyperplasia, hemimandibular elongation, and a hybrid form of UCH [1]. UCH is the most common growth disorder of the Temporo-Mandibular Joint (TMJ), yet the exact aetiology of UCH remains unclear [2].

The discrepancy in growth activity can be shown with Single Photon Emission Computed Tomography (SPECT). Saridin et al. described that a difference of more than 10% between both condyles can be seen as a significant growth differential between the left and right. This can possibly be used as a cut-off point to determine if surgery, i.e., a condylectomy, is needed [3].

One of the treatment options is performing a high condylectomy and orthognathic surgery concurrently. This can lead to good aesthetic and functional results [4]. Another possibility is performing a condylectomy, thus avoiding the need for secondary orthognathic surgery. In a high condylectomy, the most proximal part (at least 5 mm) of the mandibular condyle is removed surgically. This will stop further growth on the affected side, but the related remodelling of the facial asymmetry and occlusion reassurance is not predictable [5,6]. The height of the condyle removed during a proportional condylectomy depends on the asymmetry, i.e., a larger discrepancy (e.g., 8 mm) in ramus height means more (e.g., 8 mm) of the affected condyle will be removed. A proportional condylectomy significantly reduces the need for secondary surgery compared with a high condylectomy (15.8% vs. 90.9%) [7].

A recent systematic review also showed a tendency towards a proportional approach to avoid secondary corrective orthognathic surgery [8]. Hence, our clinic prefers initially performing only a proportional condylectomy on active UCH; the preoperative measurement of the ramus height difference has to be performed in an exact and reproducible way.

A great variety of measuring methods exist in the current literature to determine the vertical difference between the left and right ramus heights [9,10,11,12,13,14]. Previously, we used the conventional lateral transpharyngeal contact radiography method described by Parma [15] to determine the amount of condyle to resect. These radiographs are no longer available for daily clinical practice, so we determine the resection amount from panorex images. This is carried out ‘free-hand’, meaning a point is selected manually where the surgeon thinks the gonial angle (Go) is located. This point is connected to the highest point on the top of the condyle (Co). The distance in millimetres between these points is considered the ramus height. Both sides are measured, and the difference between the left and right, i.e., the amount to resect during the condylectomy, is determined. A problem encountered in our clinic with this method is that the TMJ surgeons measure different amounts to resect during surgery since this is not a validated procedure. Another commonly used measuring method for ramus height with the panorex is the ‘bisection method’, in which point Go is constructed by bisecting the angle between the tangent of the lower and posterior borders [11,12,13,16].

Unfortunately, such two-dimensional measurements on a panorex have been reported as leading to asymmetry under-diagnosis due to the angled projection of the panorex [17]. Hence, alternative planning tools need to be explored to determine the asymmetry more precisely. De Bont et al. already presented Computed Tomography (CT) in 1993 as an imaging modality to detect UCH [18]. Nolte et al. (2016) showed that measurements based on three-dimensional data, using a Cone-Beam Computed Tomography (CBCT) scan, can be used to quantify mandibular asymmetry [19]. However, the different two-dimensional panorex measurements have not been compared with each other or compared with a 3D analysis.

We asked ourselves if it would be possible to add more reproducibility, i.e., precision or reliability, to our daily practice by changing to a different measuring method. Hence, the aim of this study was to objectify the most reproducible ramus height measurement method, so that it does not matter who performs the measuring, or when. A 3D analysis method based on CBCT data was developed and compared to the panorex-free-hand method, and to the commonly used bisection method in the literature.

## 2. Materials and Methods

In order to evaluate the reproducibility of the 3 described measuring methods, we selected a cohort of patients who had had both panorex (Planmeca, Helsinki, Finland) and CBCT (Planmeca, Helsinki, Finland) scans within a short period of time: the pre-operative data of the patients who had undergone orthognathic surgery between 2015 and 2018 were analysed. Patients were only included if they met the following criteria:Older than 16 years of age;A panorex image where the condyle and the gonial angle are visible on both sides;A CBCT with a slice thickness of 0.4 mm.

The exclusion criterion was:Prior mandibular surgery.

### 2.1. Two-Dimensional Methods

Two different independent measurements of the mandibular ramus height were performed on the 2D panorex images. The first method was conducted ‘free-hand’, meaning the observer manually chose a point where he/she thought the gonial angle (Go) was located. This point was connected to the highest point on the top of the condyle (Co). The distance in millimetres between these points was considered the ramus height. Both sides were measured, and the difference between the left and right was determined. The second ‘bisection method’ (Figure 1) was based on the method described by Kjellberg et al. (1994) [13]. First, the tangent of both the mandibular ramus and the body was drawn. Another line was drawn from the intersection of these lines to the mandible, dividing the angle between the two tangents into two equal angles. The gonion, where this line crosses the curvature of the angle of the mandible, was marked. The ramus height was measured from the Go-Co on both sides, and the difference between the left and right was determined. A positive number meant the left ramus was longer compared to the right and vice versa. Both 2D measurements (bisection and free-hand method) were made for all the patients by N.B.v.B., an Oral and Maxillofacial surgeon who specialized in TMJ surgery.

### 2.2. Three-Dimensional Method

All the 3D measurements were made on segmented mandibular bones based on the CBCTs in a semi-automated way. The right side of the mandible was mirrored using Materialise ProPlan CMF 3.0 (Materialise, Leuven, Belgium), after which, it was superimposed onto the original segmentation of the left gonial angle. This area was manually selected, as shown in Figure 2. Once the observer was satisfied with the alignment, both 3D ramus shapes were exported to a standalone application created using the MATLAB R2017a (Mathworks, Natick, MA, USA) AppDesigner module. Then, the tangents of the mandibular ramus of both sides were positioned vertically, and the highest point on each condyle was marked. The difference in condylar height was equal to the vertical height difference of both points (Figure 3). All three 3D steps, i.e., segmentations, superimpositioning, and measurements, were carried out for all the patients by J.W.v.d.G., a technical physician and engineer with software experience.

### 2.3. Sample Size

A pilot study with ten randomly selected patients from the orthognathic dataset was undertaken to calculate the required sample size for drawing founded conclusions when making comparisons between the 2D methods and the 3D method. The results, i.e., the mean difference between the left and right ramus height and standard deviation (data not shown), were entered in G*Power (free available Statistical Software for Power Analysis by Department of Psychology, Dusseldorf, Germany) to calculate the effect size where *p* < 0.05 and there is an acceptable power of 0.8. The required sample size for this study was determined as 32 cases. The 32 patients were randomly selected from the orthognathic dataset using the same inclusion and exclusion criteria. To test the intra- and inter-observer reliability, 10 cases were randomly selected from the group of 32. These ten cases were re-evaluated 2 weeks after the first measurements by the same observers (N.B.v.B. (2D) and J.W.v.d.G. (3D)) for intra-observer reliability. The same cases were also evaluated by different observers (F.K.L.S., an Oral and Maxillofacial surgeon specialized in TMJ surgery (2D) and J.K., a technical physician and engineer with software experience (3D)), and their measurements were compared with those made by N.B.v.B. (2D) and J.W.v.d.G. (3D) to check for inter-observer reliability. The 3D inter-observer reliability analysis involved observer J.K. performing only the last two 3D steps (see three-dimensional method above) and J.W.v.d.G. performing the segmentations.

### 2.4. Statistical Analysis

Statistical analyses were performed with IBM SPSS Statistics 23 (SPSS, Chicago, IL, USA). To determine if the measurements were significantly different, a paired samples *t*-test was applied. *p*-values < 0.05 were considered statistically significant. The distribution of the data was checked by constructing Q-Q plots and by performing the Kolmogorov–Smirnov test [20]. To assess the intra- and inter-observer reliability, the Intraclass Correlation Coefficients (ICC; two-way mixed effects model, single measures, absolute agreement) and the 95% confidence intervals (CI) were calculated for all 3 methods. Values less than 0.5, between 0.5 and 0.75, between 0.75 and 0.9, and larger than 0.90 were indicative of poor, moderate, good, and excellent reliability, respectively [21,22]. Bland–Altman plots were constructed to analyse measurement differences between either the observers or the repeated measurements with all three methods. The ICC results were compared; our clinically acceptable difference was 1 mm [23,24].

## 3. Results

The measured population (n = 32) had an average age (±s.d.) of 26.9 (±9.6) years. The cohort was made up of 59.4% (n = 19) female and 40.6% (n = 13) male patients. The average age (±s.d.) of the ten randomly selected patients for the ICC measurements was 26.5 (±7.4) years, of which 40% (n = 4) were female and 60% (n = 6) were male.

All data had a normal distribution according to the Q-Q plots and Kolmogorov–Smirnov tests (data not shown). The panorex-free-hand method showed an average difference of 2.15 mm ± 3.53 mm between the left and right ramus heights. The panorex-bisection method showed a difference of 0.93 mm ± 3.34 mm, and the CBCT measurements showed a difference of 1.41 mm ± 2.50 mm. The average absolute difference between both 2D methods was significant, 1.40 mm ± 1.10 mm (*p* = 0.001). Additionally, the panorex-bisection and CBCT measurements showed significant average differences of 1.70 mm ± 1.17 mm (*p* = 0.001). The average difference between the panorex-free-hand and the CBCT method was 1.48 mm ± 1.13 mm, which is not a significant discrepancy (*p* = 0.25).

The average differences between the first and second measurements (same observer, 2 weeks apart) was 0.85 mm ± 0.50 mm with an intra-observer reliability of 0.95 (95% CI: 0.82–0.99) for the panorex-free-hand method. Regarding the panorex-bisection, the average difference was 0.65 mm ± 0.58 mm with an intra-observer reliability of 0.95 (95% CI: 0.82–0.99). The average difference in the CBCT measurement was 0.56 mm ± 0.39 mm with an intra-observer reliability of 0.92 (95% CI: 0.73–0.98). Appendix A shows the Bland–Altman plots of all the 2D and 3D measurements.

The average measurement differences between the observers were 1.53 mm ± 0.87 mm, 0.72 mm ± 0.37 mm, and 0.76 mm ± 0.58 mm for the free-hand, the bisection, and CBCT methods, respectively (Table 1). The ICC of the inter-observer reliability of the free-hand method was 0.86 (95% CI: 0.06–0.97), the bisection method was 0.96 (95% CI: 0.78–0.99), and the CBCT method was 0.87 (95% CI: 0.56–0.97).

## 4. Discussion

Objective and reproducible measurements are key when determining the amount to resect during a proportional condylectomy in patients with active unilateral condylar hyperactivity. The aim of the study was to objectify the most reproducible ramus height measurement method when determining facial asymmetry which can be used by whomever and whenever.

A significant difference was found between the panorex-bisection and 3D measurements, as well as between the panorex-bisection and panorex-free-hand method.

The intra-observer reliability was excellent for all three methods (ICC > 0.9). The panorex-bisection also showed excellent inter-observer reliability (ICC > 0.9), while both the CBCT and panorex-free-hand gave good results (0.75 < ICC < 0.9). However, the lower boundary of the 95% CI (0.06–0.97) of the panorex-free-hand meant the inter-observer reliability was poor. Furthermore, the average difference between the inter-observer measurements of the panorex-free-hand was 1.53 mm ± 0.87 mm, which exceeds our clinically accepted margin of 1 mm. The combination of a poor lower boundary of the 95% CI of the inter-observer reliability and exceeding a clinically acceptable margin of 1 mm suggests the panorex-free-hand method is inferior for clinical use in terms of reproducible measurements of ramus height differences.

Both the 3D and bisection methods demonstrated excellent intra-observer reliability. The 3D method had a good inter-observer ICC with a moderate–excellent 95% CI, and the bisection method had an excellent inter-observer ICC with a good–excellent 95% CI. Therefore, the bisection method seems more suitable for determining mandibular ramus height differences compared to the 2D free-hand and 3D methods. Nevertheless, the most accurate display of actual asymmetry is still undetermined because there is no gold-standard, and the precise difference between the left and right ramus heights is unknown. Preferably, consecutive measurements over time, for example, in the case of a wait-and-see policy for a possibly extinguished UCH (i.e., anamnestic increasing asymmetry, but <10% difference in activity between the condyles on a SPECT image), should be performed with the bisection method because of the excellent intra-observer reliability.

Kambylafkas et al. showed that although a panorex can be used to evaluate mandibular asymmetry, some under-diagnoses will occur [17]. The panorex projection angle of the mandibular ramus in an asymmetrical mandible could differ on both sides, possibly resulting in under-diagnoses. Moreover, the position of the head of the patient while making a panorex could affect the measured asymmetry. According to Vasudeva et al. (2012), the appearance of the mandibular condyle depends on the projection angle which relates to the head’s position [25]. This could negatively influence the measurements. To date, no research has reported on the influence of the projection angle on ramus height measurements. However, one could assume these factors are related and therefore the ramus height will be affected when a panorex is made at a different angle, especially in patients with UCH where the dental plane is often tilted. The patient has to bite on a piece of plastic (to position the head correctly before and during panorex-scanning), which could change the projection angle. This effect needs to be kept in mind when creating and evaluating the panorex. We hypothesize that, although our measurements were performed on a non-UCH group of patients, this will not have influenced the results of our study because reproducibility was the primary goal.

J.W.v.d.G. was the only observer who performed the CBCTs segmentations. Moerenhout et al. (2009) achieved excellent reliability on segmenting the CBCT using the Materialise software [26]. Although they used a different software, ours was also a CE-certified medical processing software (Proplan CMF), and we achieved excellent intra-observer reliability. We therefore deemed it unnecessary to repeat this step by observer J.K. to determine the inter-observer reliability.

Markic et al. (2015) were also able to make reliable ramus height measurements on panorex and CBCT images [9] and found excellent intra- and inter-observer reliabilities for both imaging modalities, indicating they can be used to measure asymmetry. Their measuring method was slightly different to ours as both Co and Go were constructed in a different way: Co was the intersection point of the tangent with the condyle of a line perpendicular to the tangent of the posterior border of the mandible. Hence, the Co was not the most cranial point of the condyle, but lower and more posterior. We are not sure if it is possible to correct for this when performing a proportional condylectomy, i.e., resectioning the most cranial part of the condyle. We consider this to be a tricky situation, especially when the condyle is greatly inclined anteriorly. Markic et al.’s Go point was the intersection point of the lower border of the line through Co parallel to the tangent of the posterior border of the mandible. We hypothesize that with an increasing high mandibular plane angle, and/or as the inclination of the condyle increases, the Go point will be located more anteriorly. Furthermore, their sample size was smaller, no power analyses were performed, and a 95% CI was not reported. Our Go point was the same as that described by Gaufield: a point on the curvature of the angle of the mandible located by bisecting the angle formed by lines tangent to the posterior ramus and the inferior border of the mandible [16].

Nolte et al. performed a 3D quantification of mandibular asymmetry in 37 UCH patients and compared this with a group of healthy subjects, matched for age and gender. It is unclear why they had this number of subjects. They concluded that CBCT is a useful and accurate modality for this purpose [19]. Although they performed linear measurements on the data, they did not make any comparisons with 2D methods. In another study of patients with unilateral condylar hyperplasia, Nolte et al. performed measurements on panoramic radiographs [27]. They subdivided their measurements into condylar head, condylar neck, ramus height, angle of gonion, and body height. They defined the ramus height as “the total length between the most upper and lower points perpendicular to the tangential line of the mandibular ramus”. It is not completely clear to us how this was carried out. Nevertheless, the biggest difference in between-observer reproducibility was observed for the condylar head and condylar neck measurements. Regarding the ramus height, they found a difference between the affected and healthy side: the between-observer reproducibility (kappa), as assessed on an orthopantomogram, was 0.88 for the affected side and 0.96 for the healthy side. A power analysis was not performed by that study, and therefore, firm conclusions cannot be drawn.

Sembronio et al. also performed 3D virtual mirroring by superimposing the contralateral healthy side on the condylar hyperplasia side [28]. A custom-designed condylar cutting guide was modelled on the condylar head, allowing for the precise tracing of the osteotomy as planned. This technique proved to be very useful for the seven patients treated in this way. However, the paper did not give any information about the accuracy of the planned and performed condylectomy. Combining the most reproducible and the most accurate measuring method with the guided surgery, as described by Sembronio et al., is potentially a suitable method for correcting asymmetry and should be part of future studies.

A substantial difference of 1.70 mm ± 1.17 mm was found by us between the panorex-bisection and 3D measurements, which means there was a clinically relevant (>1 mm) discrepancy between them. All the patients with more than a two-millimetre difference between measurements (which was the case with 10 of the 32 cases in total) were closely reviewed. No explanation could be found for three of the ten cases. In the other seven cases, it was difficult to identify the mandibular angle, the highest point of the condyle, or both, on the panorex image because of overprojection with other structures. Nevertheless, the intra- and inter-observer reliabilities of the panorex-bisection method were both higher compared to the 3D measurements. This indicates that the repeatability of the measurements is better, but that the accuracy of the measurements compared to the actual asymmetry is questionable in the presence of overprojection, making it difficult to identify the mandibular angle and/or the top of the condyle.

To the best of our knowledge, comparisons of different measuring methods for ramus height involving power analyses, as was performed in our study, has never been described in the available literature. This research provides a better understanding of (1) the reliability of the currently available and easily accessible 2D methods, resulting in us switching from the panorex-free-hand to the panorex-bisection method in daily practice, and (2) the possible contribution of 3D to proportional condylectomy surgery, which is promising considering the 3D method is still in an early phase of development. The most accurate display of the actual asymmetry remains undetermined because there is no gold-standard and because the precise difference between left and right ramus heights is unknown. More research needs to be carried out to determine this, including developing an easy, quick-to-use, and more reliable method for daily practice based on (CB)CTs, e.g., the use of fully automatic (1) mandible segmentations, (2) superimposition algorithms, and (3) ramus height difference measurements.

In conclusion, we discourage the use of the panorex-free-hand method in clinical use for reproducibility measurements of ramus height differences. The two-dimensional panorex-bisection method is the most reliable method, provided that the panorex images are good quality.

## Figures and Tables

**Figure 1 jpm-12-01181-f001:**
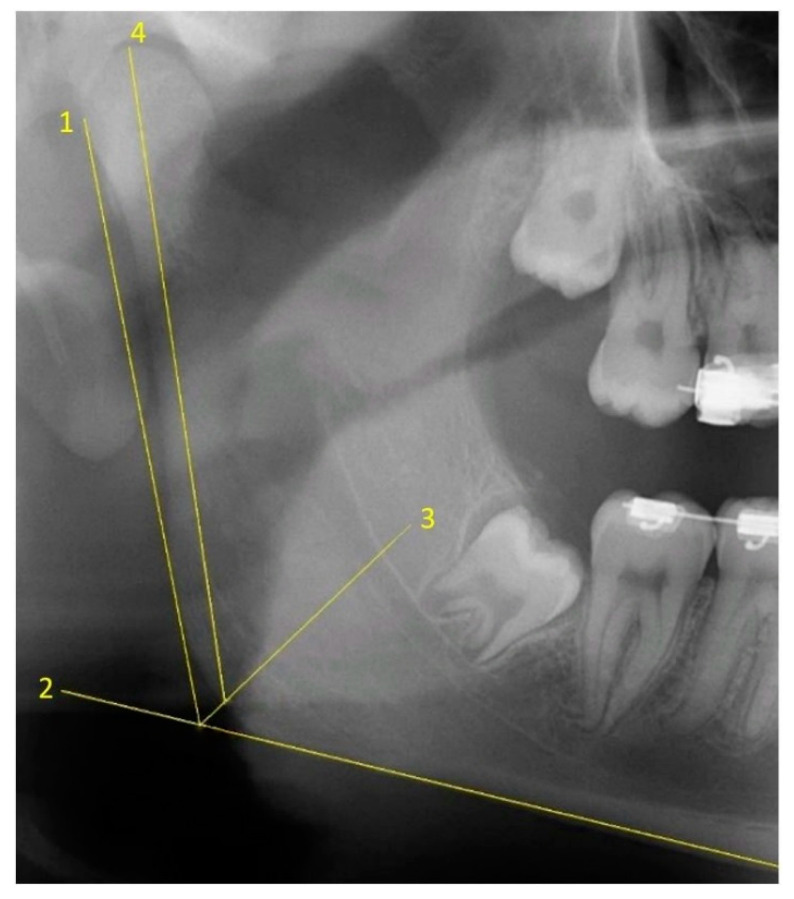
Ramus height measurement on the right side of the panorex using the bisection method. Lines 1 and 2 are the tangents of the mandibular ramus and the body, respectively. Line 3 is the bisection line dividing the angle between the two tangents in half. Line 4 is used to measure the ramus height and goes from the gonial angle (where line 3 crosses the curvature of the angle of the mandible, i.e., point gonion) to the highest point on the top of the condyle, i.e., point condyle.

**Figure 2 jpm-12-01181-f002:**
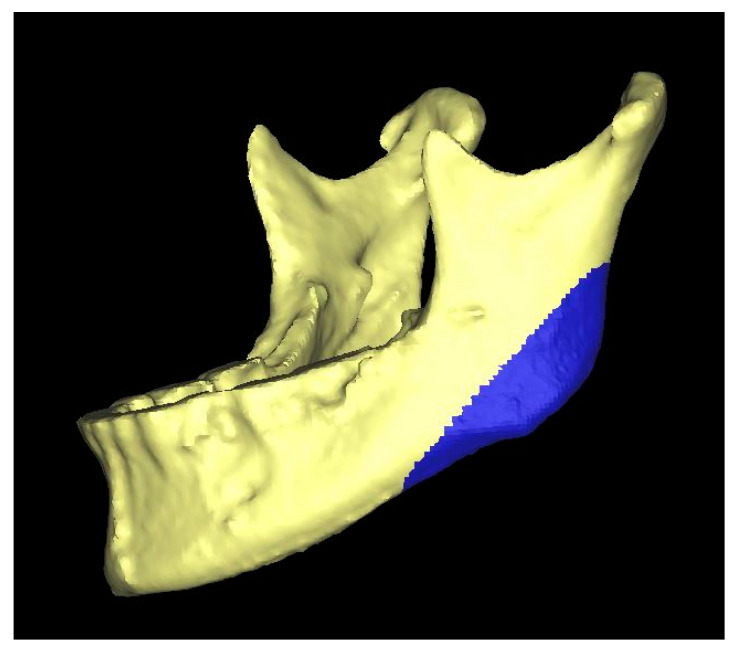
Three-dimensional representation of the mandibular bone. The blue surface is the area which was selected in order to align (i.e., superimpose) the original (left) and mirrored (right) mandibular angle.

**Figure 3 jpm-12-01181-f003:**
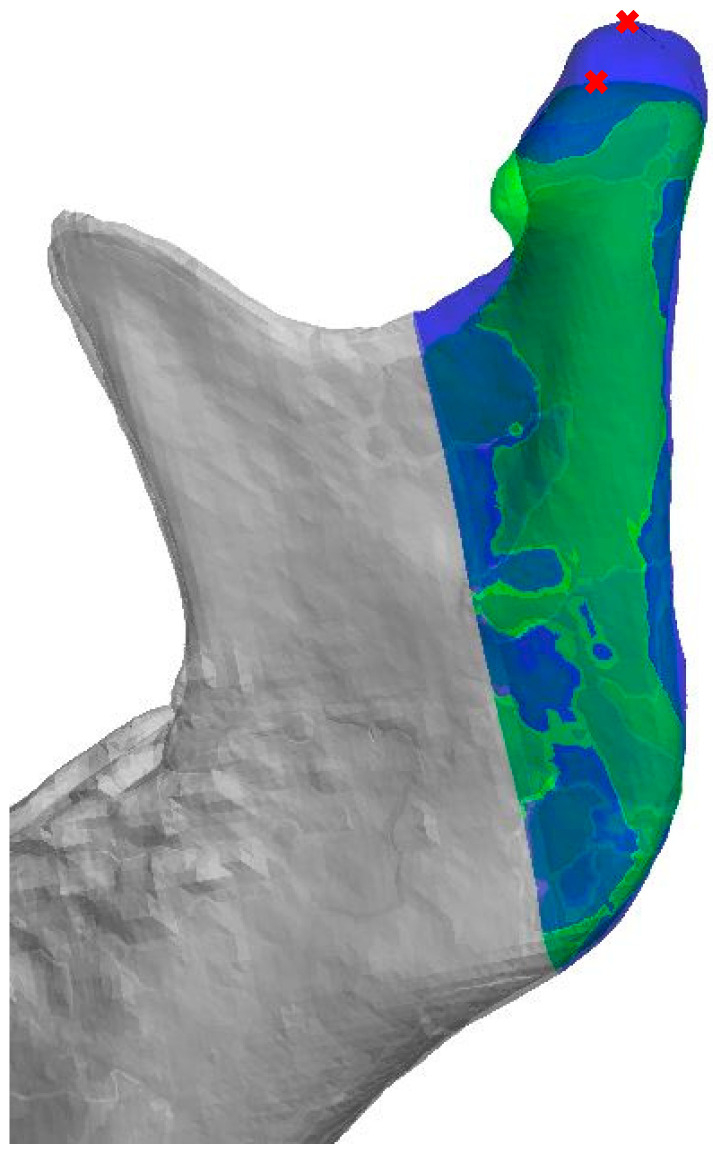
Three-dimensional representation of both the original (blue) as well as the aligned mirrored (green) side of the mandibular bone. The tangent of the mandibular ramus of both sides is positioned vertically, and the highest point on each condyle is marked in red. The difference in condylar height is equal to the vertical height difference of both points.

**Table 1 jpm-12-01181-t001:** Measuring results and reliability results.

Difference in Ramus Height Left vs. Right N = 32 *		Intra-Observer Reliability N = 10 †		Inter-Observer Reliability N = 10 ‡	
Method	Mean ± SD	Mean diff ± SD	ICC (95% CI)	Mean diff ± SD	ICC (95% CI)
OPG-FH	2.15 ± 3.53	0.85 ± 0.50	0.95 (0.82–0.99)	1.53 ± 0.87	0.86 (0.06–0.97)
OPG-B	0.93 ± 3.34	0.65 ± 0.58	0.95 (0.82–0.99)	0.72 ± 0.37	0.96 (0.78–0.99)
CBCT	1.41 ± 2.50	0.56 ± 0.39	0.92 (0.73–0.98)	0.76 ± 0.58	0.87 (0.56–0.97)

* Observer N.B.v.B. performed both the OPG-FH and the OPG-B method; observer J.W.v.d.G. performed the CBCT method. † Measurements were performed 2 weeks apart by the same observers: N.B.v.B. performed both the OPG-FH and the OPG-B method; J.W.v.d.G. performed the CBCT method. ‡ Measurements performed by different observers: N.B.v.B. vs. F.K.L.S. for the OPG-FH and OPG-B method; J.W.v.d.G. vs. J.K. for the CBCT method. Abbreviations: OPG-B = two-dimensional panorex-bisection, OPG-FH = two-dimensional panorex-free-hand, CBCT = three-dimensional mirror method.

## Data Availability

The data described in this study are available in Table 1, Figure 1, Figure 2 and Figure 3 and Appendix A. The raw data presented in this study are available on request from the corresponding author. Requests for materials should be addressed to N.B.v.B.

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
