# Peer review of "Reproducibility of 2D and 3D Ramus Height Measurements in Facial Asymmetry"

_jpm, 2022, doi:10.3390/jpm12071181_

Round 1
Reviewer 1 Report
Thank you for giving me the chance to review this interesting work. I evaluated the study "Reproducibility of 2D and 3D ramus height measurements in 2 facial asymmetry", and I have some concerns that I want to share with the authors.
In general, the article is written in a very understandable and straightforward language.
The first three paragraphs of the discussion section contain results. I think that there is too much emphasis on inter-observer reliability.
I recommend adding the limitations of trying to the discussion.
Sometimes the changes in the bone may not have the same effect on the soft tissue. Have the clinical results and patient satisfaction been evaluated for the three different methods in the article?
Best regards.
Reviewer 2 Report
This study aimed to determine the most reproducible ramus height measuring method, comparing two 2D and one 3D methods.
Overall, the paper is well written, but three aspects of the Methods must be clarified:
1. Be more precise about how you estimated the sample size, which parameters and values you used?
2. It is not clear what was the distribution of the data.
3. Please describe the raters who participated in the study and what were the criteria to select them.
